# Convolutional Neural Network Architectures for Matching Natural Language Sentences

**Baotian Hu**[§*]    **Zhengdong Lu**[†]    **Hang Li**[†]    **Qingcai Chen**[§]

[§]Department of Computer Science
& Technology, Harbin Institute of Technology
Shenzhen Graduate School, Xili, China
`baotianchina@gmail.com`
`qingcai.chen@hitsz.edu.cn`

[†] Noah's Ark Lab
Huawei Technologies Co. Ltd.
Sha Tin, Hong Kong
`lu.zhengdong@huawei.com`
`hangli.hl@huawei.com`

## Abstract

Semantic matching is of central importance to many natural language tasks [2, 28]. A successful matching algorithm needs to adequately model the internal structures of language objects and the interaction between them. As a step toward this goal, we propose convolutional neural network models for matching two sentences, by adapting the convolutional strategy in vision and speech. The proposed models not only nicely represent the hierarchical structures of sentences with their layer-by-layer composition and pooling, but also capture the rich matching patterns at different levels. Our models are rather generic, requiring no prior knowledge on language, and can hence be applied to matching tasks of different nature and in different languages. The empirical study on a variety of matching tasks demonstrates the efficacy of the proposed model on a variety of matching tasks and its superiority to competitor models.

## 1 Introduction

Matching two potentially heterogenous language objects is central to many natural language applications [28, 2]. It generalizes the conventional notion of similarity (e.g., in paraphrase identification [19]) or relevance (e.g., in information retrieval[27]), since it aims to model the correspondence between "linguistic objects" of different nature at different levels of abstractions. Examples include top-$k$ re-ranking in machine translation (e.g., comparing the meanings of a French sentence and an English sentence [5]) and dialogue (e.g., evaluating the appropriateness of a response to a given utterance[26]).

Natural language sentences have complicated structures, both sequential and hierarchical, that are essential for understanding them. A successful sentence-matching algorithm therefore needs to capture not only the internal structures of sentences but also the rich patterns in their interactions. Towards this end, we propose deep neural network models, which adapt the convolutional strategy (proven successful on image [11] and speech [1]) to natural language. To further explore the relation between representing sentences and matching them, we devise a novel model that can naturally host both the hierarchical composition for sentences and the simple-to-comprehensive fusion of matching patterns with the same convolutional architecture. Our model is generic, requiring no prior knowledge of natural language (e.g., parse tree) and putting essentially no constraints on the matching tasks. This is part of our continuing effort[1] in understanding natural language objects and the matching between them [13, 26].

---

[*]The work is done when the first author worked as intern at Noah's Ark Lab, Huawei Techologies
[1]Our project page: `http://www.noahlab.com.hk/technology/Learning2Match.html`

Our main contributions can be summarized as follows. First, we devise novel deep convolutional network architectures that can naturally combine 1) the hierarchical sentence modeling through layer-by-layer composition and pooling, and 2) the capturing of the rich matching patterns at different levels of abstraction; Second, we perform extensive empirical study on tasks with different scales and characteristics, and demonstrate the superior power of the proposed architectures over competitor methods.

**Roadmap**   We start by introducing a convolution network in Section 2 as the basic architecture for sentence modeling, and how it is related to existing sentence models. Based on that, in Section 3, we propose two architectures for sentence matching, with a detailed discussion of their relation. In Section 4, we briefly discuss the learning of the proposed architectures. Then in Section 5, we report our empirical study, followed by a brief discussion of related work in Section 6.

## 2   Convolutional Sentence Model

We start with proposing a new convolutional architecture for modeling sentences. As illustrated in Figure 1, it takes as input the embedding of words (often trained beforehand with unsupervised methods) in the sentence aligned sequentially, and summarize the meaning of a sentence through layers of convolution and pooling, until reaching a fixed length vectorial representation in the final layer. As in most convolutional models [11, 1], we use convolution units with a local "receptive field" and shared weights, but we design a large feature map to adequately model the rich structures in the composition of words.

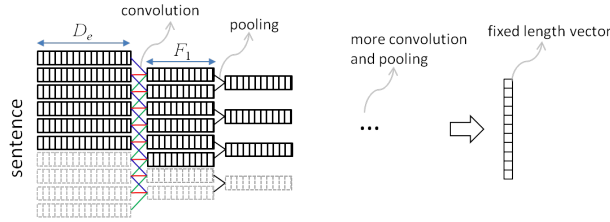

Figure 1: The over all architecture of the convolutional sentence model. A box with dashed lines indicates all-zero padding turned off by the gating function (see top of Page 3).

**Convolution**   As shown in Figure 1, the convolution in Layer-1 operates on sliding windows of words (width $k_1$), and the convolutions in deeper layers are defined in a similar way. Generally,with sentence input $\mathbf{x}$, the convolution unit for feature map of type-$f$ (among $F_\ell$ of them) on Layer-$\ell$ is

$$z_i^{(\ell,f)} \overset{\text{def}}{=} z_i^{(\ell,f)}(\mathbf{x}) = \sigma(\mathbf{w}^{(\ell,f)}\hat{\mathbf{z}}_i^{(\ell-1)} + b^{(\ell,f)}), \quad f = 1, 2, \cdots, F_\ell \tag{1}$$

and its matrix form is $\mathbf{z}_i^{(\ell)} \overset{\text{def}}{=} \mathbf{z}_i^{(\ell)}(\mathbf{x}) = \sigma(\mathbf{W}^{(\ell)}\hat{\mathbf{z}}_i^{(\ell-1)} + \mathbf{b}^{(\ell)})$, where

- $z_i^{(\ell,f)}(\mathbf{x})$ gives the output of feature map of type-$f$ for location $i$ in Layer-$\ell$;

- $\mathbf{w}^{(\ell,f)}$ is the parameters for $f$ on Layer-$\ell$, with matrix form $\mathbf{W}^{(\ell)} \overset{\text{def}}{=} [\mathbf{w}^{(\ell,1)}, \cdots, \mathbf{w}^{(\ell,F_\ell)}]$;

- $\sigma(\cdot)$ is the activation function (e.g., Sigmoid or Relu [7])

- $\hat{\mathbf{z}}_i^{(\ell-1)}$ denotes the segment of Layer-$\ell$−1 for the convolution at location $i$ , while

$$\hat{\mathbf{z}}_i^{(0)} = \mathbf{x}_{i:i+k_1-1} \overset{\text{def}}{=} [\mathbf{x}_i^\top, \ \mathbf{x}_{i+1}^\top, \ \cdots, \ \mathbf{x}_{i+k_1-1}^\top]^\top$$

concatenates the vectors for $k_1$ (width of sliding window) words from sentence input $\mathbf{x}$.

**Max-Pooling**   We take a max-pooling in every two-unit window for every $f$, after each convolution

$$z_i^{(\ell,f)} = \max(z_{2i-1}^{(\ell-1,f)}, z_{2i}^{(\ell-1,f)}), \quad \ell = 2, 4, \cdots .$$

The effects of pooling are two-fold: 1) it shrinks the size of the representation by half, thus quickly absorbs the differences in length for sentence representation, and 2) it filters out undesirable composition of words (see Section 2.1 for some analysis).

**Length Variability**    The variable length of sentences in a fairly broad range can be readily handled with the convolution and pooling strategy. More specifically, we put all-zero padding vectors after the last word of the sentence until the maximum length. To eliminate the boundary effect caused by the great variability of sentence lengths, we add to the convolutional unit a gate which sets the output vectors to all-zeros if the input is all zeros. For any given sentence input $\mathbf{x}$, the output of type-$f$ filter for location $i$ in the $\ell^{th}$ layer is given by

$$z_i^{(\ell,f)} \stackrel{\text{def}}{=} z_i^{(\ell,f)}(\mathbf{x}) = g(\hat{\mathbf{z}}_i^{(\ell-1)}) \cdot \sigma(\mathbf{w}^{(\ell,f)}\hat{\mathbf{z}}_i^{(\ell-1)} + b^{(\ell,f)}), \tag{2}$$

where $g(\mathbf{v}) = 0$ if all the elements in vector $\mathbf{v}$ equals 0, otherwise $g(\mathbf{v}) = 1$. This gate, working with max-pooling and positive activation function (e.g., Sigmoid), keeps away the artifacts from padding in all layers. Actually it creates a natural hierarchy of all-zero padding (as illustrated in Figure 1), consisting of nodes in the neural net that would not contribute in the forward process (as in prediction) and backward propagation (as in learning).

## 2.1   Some Analysis on the Convolutional Architecture

The convolutional unit, when combined with max-pooling, can act as the compositional operator with local selection mechanism as in the recursive autoencoder [21]. Figure 2 gives an example on what could happen on the first two layers with input sentence "The cat sat on the mat". Just for illustration purpose, we present a dramatic choice of parameters (by turning off some elements in $\mathbf{W}^{(1)}$) to make the convolution units focus on different segments within a 3-word window. For example, some feature maps (group 2) give compositions for "the cat"

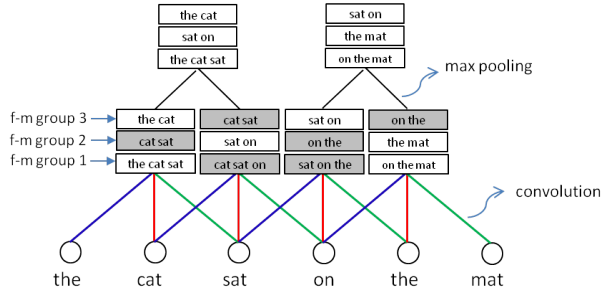

Figure 2: The cat example, where in the convolution layer, gray color indicates less confidence in composition.

and "cat sat", each being a vector. Different feature maps offer a variety of compositions, with confidence encoded in the values (color coded in output of convolution layer in Figure 2). The pooling then chooses, *for each composition type*, between two adjacent sliding windows, e.g., between "on the" and "the mat" for feature maps group 2 from the rightmost two sliding windows.

**Relation to Recursive Models**    Our convolutional model differs from Recurrent Neural Network (RNN, [15]) and Recursive Auto-Encoder (RAE, [21]) in several important ways. First, unlike RAE, it does not take a single path of word/phrase composition determined either by a separate gating function [21], an external parser [19], or just natural sequential order [20]. Instead, it takes multiple choices of composition via a large feature map (encoded in $\mathbf{w}^{(\ell,f)}$ for different $f$), and leaves the choices to the pooling afterwards to pick the more appropriate segments(in every adjacent two) for each composition. With any window width $k_\ell \geq 3$, the type of composition would be much richer than that of RAE. Second, our convolutional model can take supervised training and tune the parameters for a specific task, a property vital to our supervised learning-to-match framework. However, unlike recursive models [20, 21], the convolutional architecture has a fixed depth, which bounds the level of composition it could do. For tasks like matching, this limitation can be largely compensated with a network afterwards that can take a "global" synthesis on the learned sentence representation.

**Relation to "Shallow" Convolutional Models**    The proposed convolutional sentence model takes simple architectures such as [18, 10] (essentially the same convolutional architecture as SENNA [6]), which consists of a convolution layer and a max-pooling over the entire sentence for each feature map. This type of models, with local convolutions and a global pooling, essentially do a "soft" local template matching and is able to detect local features useful for a certain task. Since the sentence-level sequential order is inevitably lost in the global pooling, the model is incapable of modeling more complicated structures. It is not hard to see that our convolutional model degenerates to the SENNA-type architecture if we limit the number of layers to be two and set the pooling window infinitely large.

# 3   Convolutional Matching Models

Based on the discussion in Section 2, we propose two related convolutional architectures, namely ARC-I and ARC-II), for matching two sentences.

## 3.1   Architecture-I (ARC-I)

Architecture-I (ARC-I), as illustrated in Figure 3, takes a conventional approach: It first finds the representation of each sentence, and then compares the representation for the two sentences with a multi-layer perceptron (MLP) [3]. It is essentially the Siamese architecture introduced in [2, 11], which has been applied to different tasks as a nonlinear similarity function [23]. Although ARC-I enjoys the flexibility brought by the convolutional sentence model, it suffers from a drawback inherited from the Siamese architecture: it defers the interaction between two sentences (in the final MLP) to until their individual representation matures (in the convolution model), therefore runs at the risk of losing details (e.g., a city name) important for the *matching task* in representing the sentences. In other words, in the forward phase (prediction), the representation of each sentence is formed without knowledge of each other. This *cannot* be adequately circumvented in backward phase (learning), when the convolutional model learns to extract structures informative for matching on a population level.

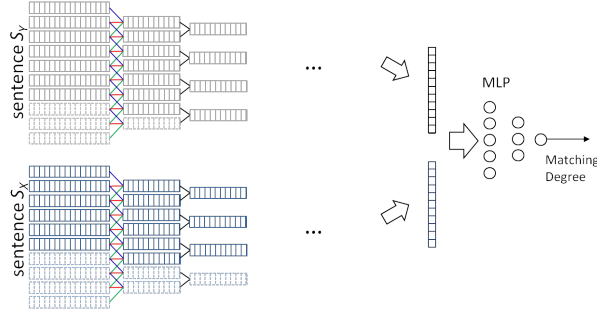

Figure 3: Architecture-I for matching two sentences.

## 3.2   Architecture-II (ARC-II)

In view of the drawback of Architecture-I, we propose Architecture-II (ARC-II) that is built directly on the interaction space between two sentences. It has the desirable property of letting two sentences meet before their own high-level representations mature, while still retaining the space for the individual development of abstraction of each sentence. Basically, in Layer-1, we take sliding windows on both sentences, and model all the *possible* combinations of them through "one-dimensional" (1D) convolutions. For segment $i$ on $S_X$ and segment $j$ on $S_Y$, we have the feature map

$$z_{i,j}^{(1,f)} \stackrel{\text{def}}{=} z_{i,j}^{(1,f)}(\mathbf{x}, \mathbf{y}) = g(\hat{\mathbf{z}}_{i,j}^{(0)}) \cdot \sigma(\mathbf{w}^{(\ell,f)}\hat{\mathbf{z}}_{i,j}^{(0)} + b^{(\ell,f)}), \tag{3}$$

where $\hat{\mathbf{z}}_{i,j}^{(0)} \in \mathbb{R}^{2k_1 D_e}$ simply concatenates the vectors for sentence segments for $S_X$ and $S_Y$:

$$\hat{\mathbf{z}}_{i,j}^{(0)} = [\mathbf{x}_{i:i+k_1-1}^{\top}, \ \mathbf{y}_{j:j+k_1-1}^{\top}]^{\top}.$$

Clearly the 1D convolution preserves the location information about both segments. After that in Layer-2, it performs a 2D max-pooling in non-overlapping $2 \times 2$ windows (illustrated in Figure 5)

$$z_{i,j}^{(2,f)} = \max(\{z_{2i-1,2j-1}^{(2,f)}, z_{2i-1,2j}^{(2,f)}, z_{2i,2j-1}^{(2,f)}, z_{2i,2j}^{(2,f)}\}). \tag{4}$$

In Layer-3, we perform a 2D convolution on $k_3 \times k_3$ windows of output from Layer-2:

$$z_{i,j}^{(3,f)} = g(\hat{\mathbf{z}}_{i,j}^{(2)}) \cdot \sigma(\mathbf{W}^{(3,f)}\hat{\mathbf{z}}_{i,j}^{(2)} + b^{(3,f)}). \tag{5}$$

This could go on for more layers of 2D convolution and 2D max-pooling, analogous to that of convolutional architecture for image input [11].

**The 2D-Convolution**   After the first convolution, we obtain a low level representation of the interaction between the two sentences, and from then we obtain a high level representation $\mathbf{z}_{i,j}^{(\ell)}$ which encodes the information from both sentences. The general two-dimensional convolution is formulated as

$$\mathbf{z}_{i,j}^{(\ell)} = g(\hat{\mathbf{z}}_{i,j}^{(\ell-1)}) \cdot \sigma(\mathbf{W}^{(\ell)}\hat{\mathbf{z}}_{i,j}^{(\ell-1)} + \mathbf{b}^{(\ell,f)}), \ \ \ell = 3, 5, \cdots \tag{6}$$

where $\hat{\mathbf{z}}_{i,j}^{(\ell-1)}$ concatenates the corresponding vectors from its 2D receptive field in Layer-$\ell$−1. This pooling has different mechanism as in the 1D case, for it selects *not only* among compositions on different segments but also among different local matchings. This pooling strategy resembles the dynamic pooling in [19] in a similarity learning context, but with two distinctions: 1) it happens on a fixed architecture and 2) it has much richer structure than just similarity.

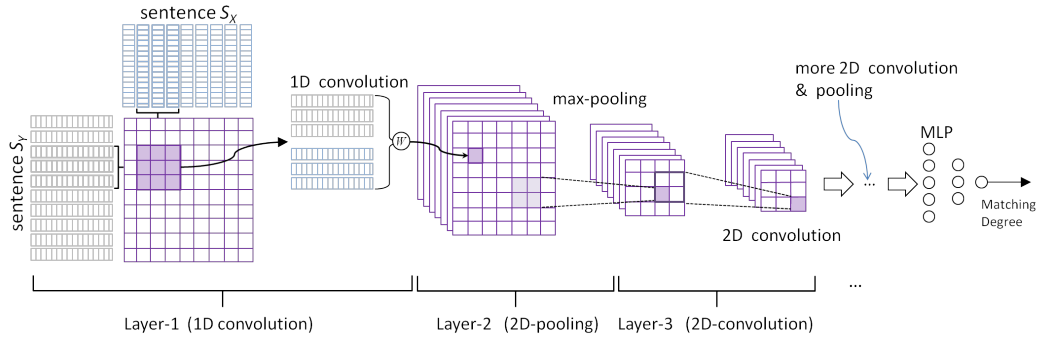

Figure 4: Architecture-II (ARC-II) of convolutional matching model

## 3.3 Some Analysis on ARC-II

**Order Preservation** Both the convolution and pooling operation in Architecture-II have this order preserving property. Generally, $\mathbf{z}_{i,j}^{(\ell)}$ contains information about the words in $S_X$ before those in $\mathbf{z}_{i+1,j}^{(\ell)}$, although they may be generated with slightly different segments in $S_Y$, due to the 2D pooling (illustrated in Figure 5). The orders is however retained in a "conditional" sense. Our experiments show that when ARC-II is trained on the $(S_X, S_Y, \tilde{S}_Y)$ triples where $\tilde{S}_Y$ randomly shuffles the words in $S_Y$, it consistently gains some ability of finding the correct $S_Y$ in the usual contrastive negative sampling setting, which however does not happen with ARC-I.

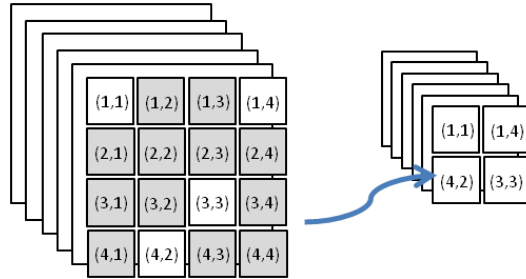

Figure 5: Order preserving in 2D-pooling.

**Model Generality** It is not hard to show that ARC-II actually subsumes ARC-I as a special case. Indeed, in ARC-II if we choose (by turning off some parameters in $\mathbf{W}^{(\ell,\cdot)}$) to keep the representations of the two sentences separated until the final MLP, ARC-II can actually act fully like ARC-I, as illustrated in Figure 6. More specifically, if we let the feature maps in the first convolution layer to be either devoted to $S_X$ or devoted to $S_Y$ (instead of taking both as in general case), the output of each segment-pair is naturally divided into two corresponding groups. As a result, the output for each filter $f$, denoted $\mathbf{z}_{1:n,1:n}^{(1,f)}$ ($n$ is the number of sliding windows), will be of rank-one, possessing essentially the same information as the result of the first convolution layer in ARC-I. Clearly the 2D pooling that follows will reduce to 1D pooling, with this separateness preserved. If we further limit the parameters in the second convolution units (more specifically $\mathbf{w}^{(2,f)}$) to those for $S_X$ and $S_Y$, we can ensure the individual development of different levels of abstraction on each side, and fully recover the functionality of ARC-I.

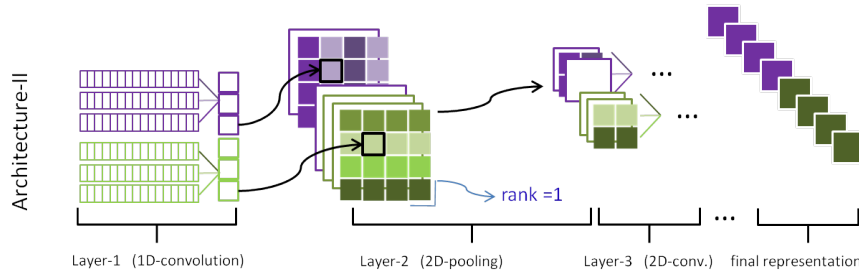

Figure 6: ARC-I as a special case of ARC-II. Better viewed in color.

As suggested by the order-preserving property and the generality of ARC-II, this architecture offers not only the capability but also the inductive bias for the individual development of internal abstraction on each sentence, despite the fact that it is built on the interaction between two sentences. As a result, ARC-II can naturally blend two seemingly diverging processes: 1) the successive composition within each sentence, and 2) the extraction and fusion of matching patterns between them, hence is powerful for matching linguistic objects with rich structures. This intuition is verified by the superior performance of ARC-II in experiments (Section 5) on different matching tasks.

## 4 Training

We employ a discriminative training strategy with a large margin objective. Suppose that we are given the following triples $(\mathbf{x}, \mathbf{y}^{+}, \mathbf{y}^{-})$ from the oracle, with $\mathbf{x}$ matched with $\mathbf{y}^{+}$ better than with $\mathbf{y}^{-}$. We have the following ranking-based loss as objective:

$$e(\mathbf{x}, \mathbf{y}^{+}, \mathbf{y}^{-}; \Theta) = \max(0, 1 + \mathbf{s}(\mathbf{x}, \mathbf{y}^{-}) - \mathbf{s}(\mathbf{x}, \mathbf{y}^{+})),$$

where $\mathbf{s}(\mathbf{x}, \mathbf{y})$ is predicted matching score for $(\mathbf{x}, \mathbf{y})$, and $\Theta$ includes the parameters for convolution layers and those for the MLP. The optimization is relatively straightforward for both architectures with the standard back-propagation. The gating function (see Section 2) can be easily adopted into the gradient by discounting the contribution from convolution units that have been turned off by the gating function. In other words, We use stochastic gradient descent for the optimization of models. All the proposed models perform better with mini-batch ($100 \sim 200$ in sizes) which can be easily parallelized on single machine with multi-cores. For regularization, we find that for both architectures, early stopping [16] is enough for models with medium size and large training sets (with over 500K instances). For small datasets (less than 10k training instances) however, we have to combine early stopping and dropout [8] to deal with the serious overfitting problem.

We use 50-dimensional word embedding trained with the Word2Vec [14]: the embedding for English words (Section 5.2 & 5.4) is learnt on Wikipedia ($\sim$1B words), while that for Chinese words (Section 5.3) is learnt on Weibo data ($\sim$300M words). Our other experiments (results omitted here) suggest that fine-tuning the word embedding can further improve the performances of all models, at the cost of longer training. We vary the maximum length of words for different tasks to cope with its longest sentence. We use 3-word window throughout all experiments[2], but test various numbers of feature maps (typically from 200 to 500), for optimal performance. ARC-II models for all tasks have eight layers (three for convolution, three for pooling, and two for MLP), while ARC-I performs better with less layers (two for convolution, two for pooling, and two for MLP) and more hidden nodes. We use ReLu [7] as the activation function for all of models (convolution and MLP), which yields comparable or better results to sigmoid-like functions, but converges faster.

## 5 Experiments

We report the performance of the proposed models on three matching tasks of different nature, and compare it with that of other competitor models. Among them, the first two tasks (namely, Sentence Completion and Tweet-Response Matching) are about matching of language objects of heterogenous natures, while the third one (paraphrase identification) is a natural example of matching homogeneous objects. Moreover, the three tasks involve two languages, different types of matching, and distinctive writing styles, proving the broad applicability of the proposed models.

### 5.1 Competitor Methods

- WORDEMBED: We first represent each short-text as the sum of the embedding of the words it contains. The matching score of two short-texts are calculated with an MLP with the embedding of the two documents as input;
- DEEPMATCH: We take the matching model in [13] and train it on our datasets with 3 hidden layers and 1,000 hidden nodes in the first hidden layer;
- URAE+MLP: We use the Unfolding Recursive Autoencoder [19][3] to get a 100-dimensional vector representation of each sentence, and put an MLP on the top as in WORDEMBED;
- SENNA+MLP/SIM: We use the SENNA-type sentence model for sentence representation;

- SenMLP: We take the whole sentence as input (with word embedding aligned sequentially), and use an MLP to obtain the score of coherence.

All the competitor models are trained on the same training set as the proposed models, and we report the best test performance over different choices of models (e.g., the number and size of hidden layers in MLP).

## 5.2 Experiment I: Sentence Completion

This is an artificial task designed to elucidate how different matching models can capture the correspondence between two clauses within a sentence. Basically, we take a sentence from Reuters [12]with two "balanced" clauses (with $8\sim 28$ words) divided by one comma, and use the first clause as $S_X$ and the second as $S_Y$. The task is then to recover the original second clause for any given first clause. The matching here is considered heterogeneous since the relation between the two is nonsymmetrical on both lexical and semantic levels. We deliberately make the task harder by using negative second clauses similar to the original ones[4], both in training and testing. One representative example is given as follows:

$S_X$: *Although the state has only four votes in the Electoral College,*

$S_Y^+$: *its loss would be a symbolic blow to republican presidential candidate Bob Dole.*

$S_Y^-$: *but it failed to garner enough votes to override an expected veto by president Clinton.*

All models are trained on 3 million triples (from 600K positive pairs), and tested on 50K positive pairs, each accompanied by four negatives, with results shown in Table 1. The two proposed models get nearly half of the cases right[5], with large margin over other sentence models and models without explicit sequence

| Model | P@1(%) |
|---|---|
| Random Guess | 20.00 |
| DeepMatch | 32.5 |
| WordEmbed | 37.63 |
| SenMLP | 36.14 |
| Senna+MLP | 41.56 |
| uRAE+MLP | 25.76 |
| Arc-I | 47.51 |
| Arc-II | **49.62** |

Table 1: Sentence Completion.

modeling. Arc-II outperforms Arc-I significantly, showing the power of joint modeling of matching and sentence meaning. As another convolutional model, Senna+MLP performs fairly well on this task, although still running behind the proposed convolutional architectures since it is too shallow to adequately model the sentence. It is a bit surprising that uRAE comes last on this task, which might be caused by the facts that 1) the representation model (including word-embedding) is not trained on Reuters, and 2) the split-sentence setting hurts the parsing, which is vital to the quality of learned sentence representation.

## 5.3 Experiment II: Matching A Response to A Tweet

We trained our model with 4.5 million original (tweet, response) pairs collected from Weibo, a major Chinese microblog service [26]. Compared to Experiment I, the writing style is obviously more free and informal. For each positive pair, we find ten random responses as negative examples, rendering 45 million triples for training. One example (translated to English) is given below, with $S_X$ standing for the tweet, $S_Y^+$ the original response, and $S_Y^-$ the randomly selected response: $S_X$: *Damn, I have to work overtime this weekend!*

$S_Y^+$: *Try to have some rest buddy.*

$S_Y^-$: *It is hard to find a job, better start polishing your resume.*

| Model | P@1(%) |
|---|---|
| Random Guess | 20.00 |
| DeepMatch | 49.85 |
| WordEmbed | 54,31 |
| SenMLP | 52.22 |
| Senna+MLP | 56.48 |
| Arc-I | 59.18 |
| Arc-II | **61.95** |

Table 2: Tweet Matching.

We hold out 300K original (tweet, response) pairs and test the matching model on their ability to pick the original response from *four* random negatives, with results reported in Table 2. This task is slightly easier than Experiment I , with more training instances and purely random negatives. It requires less about the grammatical rigor but more on detailed modeling of loose and local matching patterns (e.g., `work-overtime`⇔ `rest`). Again Arc-II beats other models with large margins, while two convolutional sentence models Arc-I and Senna+MLP come next.

### 5.4 Experiment III: Paraphrase Identification

Paraphrase identification aims to determine whether two sentences have the same meaning, a problem considered a touchstone of natural language understanding. This experiment is included to test our methods on matching homogenous objects. Here we use the benchmark MSRP dataset [17], which contains 4,076 instances for training and 1,725 for test. We use all the training instances and report the test performance from early stopping. As stated earlier, our model is not specially tailored for modeling synonymy, and generally requires $\geq 100K$ instances to work favorably. Nevertheless, our generic matching models still manage to perform reasonably well, achieving an accuracy and F1 score close to the best performer in 2008 based on hand-crafted features [17], but still significantly lower than the state-of-the-art (76.8%/83.6%), achieved with unfolding-RAE and other features designed for this task [19].

| Model | Acc. (%) | F1(%) |
|---|---|---|
| Baseline | 66.5 | 79.90 |
| Rus et al. (2008) | 70.6 | 80.5 |
| WORDEMBED | 68.7 | 80.49 |
| SENNA+MLP | 68.4 | 79.7 |
| SENMLP | 68.4 | 79.5 |
| ARC-I | 69.6 | 80.27 |
| ARC-II | 69.9 | 80.91 |

Table 3: The results on Paraphrase.

### 5.5 Discussions

ARC-II outperforms others significantly when the training instances are relatively abundant (as in Experiment I & II). Its superiority over ARC-I, however, is less salient when the sentences have deep grammatical structures and the matching relies less on the local matching patterns, as in Experiment-I. This therefore raises the interesting question about how to balance the representation of matching and the representations of objects, and whether we can guide the learning process through something like curriculum learning [4].

As another important observation, convolutional models (ARC-I & II, SENNA+MLP) perform favorably over bag-of-words models, indicating the importance of utilizing sequential structures in understanding and matching sentences. Quite interestingly, as shown by our other experiments, ARC-I and ARC-II trained purely with random negatives automatically gain some ability in telling whether the words in a given sentence are in right sequential order (with around 60% accuracy for both). It is therefore a bit surprising that an auxiliary task on identifying the correctness of word order in the response does not enhance the ability of the model on the original matching tasks.

We noticed that simple sum of embedding learned via Word2Vec [14] yields reasonably good results on all three tasks. We hypothesize that the Word2Vec embedding is trained in such a way that the vector summation can act as a simple composition, and hence retains a fair amount of meaning in the short text segment. This is in contrast with other bag-of-words models like DEEPMATCH [13].

## 6 Related Work

Matching structured objects rarely goes beyond estimating the similarity of objects in the same domain [23, 24, 19], with few exceptions like [2, 18]. When dealing with language objects, most methods still focus on seeking vectorial representations in a common latent space, and calculating the matching score with inner product[18, 25]. Few work has been done on building a deep architecture on the interaction space for texts-pairs, but it is largely based on a bag-of-words representation of text [13].

Our models are related to the long thread of work on sentence representation. Aside from the models with recursive nature [15, 21, 19] (as discussed in Section 2.1), it is fairly common practice to use the sum of word-embedding to represent a short-text, mostly for classification [22]. There is very little work on convolutional modeling of language. In addition to [6, 18], there is a very recent model on sentence representation with dynamic convolutional neural network [9]. This work relies heavily on a carefully designed pooling strategy to handle the variable length of sentence with a relatively small feature map, tailored for classification problems with modest sizes.

## 7 Conclusion

We propose deep convolutional architectures for matching natural language sentences, which can nicely combine the hierarchical modeling of individual sentences and the patterns of their matching. Empirical study shows our models can outperform competitors on a variety of matching tasks.

**Acknowledgments:** B. Hu and Q. Chen are supported in part by National Natural Science Foundation of China 61173075. Z. Lu and H. Li are supported in part by China National 973 project 2014CB340301.

## Footnotes

[2]Our other experiments suggest that the performance can be further increased with wider windows.

[3]Code from: http://nlp.stanford.edu/~socherr/classifyParaphrases.zip

[4]We select from a random set the clauses that have $0.7\sim0.8$ cosine similarity with the original. The dataset and more information can be found from http://www.noahlab.com.hk/technology/Learning2Match.html

[5]Actually Arc-II can achieve 74+% accuracy with random negatives.

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
