[Reviews · NeurIPS 2014]

Submitted by Assigned_Reviewer_1

Summary:
The paper presents two novel convolutional neural network architectures for modeling sentences in natural languages. These networks are trained specifically for the problem of matching a pair of sentences.

The first architecture is a minor modification to the standard way of using a convolutional network over natural language sentences. After a convolution operation on the word embeddings, instead of doing a pooling operation across time (full sequence of words in a sentence) to select a single feature (or k features), the proposed model applies pooling to features associated with consecutive pairs of words. This results in a output which is of half the size as the input. One can then apply multiple pairs of similar convolution-pooling layer to this output. In order to match pairs of sentences, this architecture is trained using a ranking loss applied to the final feature vector obtained from this network.

The second architecture is far more novel than the first. It also includes convolutions over sentences, however instead of applying convolutions independently to each sentence, the model applies convolutions to all the possible pairs of windows on the pair of sentences to be matched. The idea is to enable the convolution kernels to capture the local interactions among words in the two sentences at low level, as opposed to at the level when the sentence representations have matured (as in the case of the first architecture).

Both these models are evaluated on three tasks (two of them are somewhat artificial). They seem to beat the state-of-the-art (under certain conditions: discussed below).

Quality:
I think the models proposed in the paper are quite novel and certainly interesting (and useful) for others working in the same domain to try them out.

However my biggest objection is with the evaluation of the proposed models. The methodology of pre-training the word embeddings in an unsupervised way (word2vec in their case) and fixing them while training the subsequent convolutional neural network architecture is some what artificial and constrained. There is nothing stopping us from back propagating the gradients back into the word embeddings and tuning them to the task. Usually, when one does that, one observes a significant boost in performance. In particular, systems like senna (which are shown to be much worse than the proposed model in this paper) work extremely well when the embeddings are learnt for the task. My conjecture is that when the embeddings are allowed to learn for the task, architectures where pooling results in selection of a single (or k) features across the whole sentence are sufficient to capture both the internal structures of language objects and interactions among them.

In summary, the experimental results as they stand are inconclusive. We do not yet know how superior the proposed model are against the state-of-the-art, and whether we are gaining anything significant.

A few other points of note.
* While the authors cite reference [9] (which uses k-max pooling in convolutional networks and have been shown to perform quite well), they do not compare it to their model. Any reason why? [9] recently has been shown to work quite well on standard nlp tasks.

* The proposed model does not beat the state of the art in task 3 (Paraphrase Identification). It is understandable why is that the case: primarily lack of training data. However, interestingly enough, the authors do not show the numbers in Table 3. Again, any reason why?

Clarity:
The paper is reasonably well written and easy to understand. Though it is
riddled with typos which need to be fixed. An incomplete list of typos is:
* 324: model,and > model, and
* 325: the their best > the best
* 334: hetergenous > heterogeneous
* 373: requires less about > this whole sentence requires a complete re-phrase

Lastly, the details of their experimental setup were somewhat lacking as well. The values of the various model hyper-parameters used during the experiments are not clearly stated. Its also not clear whether the datasets used in experiments 1 and 2 are available for public use or not.

Originality:
The ideas proposed in the paper are fairly original and interesting. Subject to their correct evaluation and validation, I think they can be very useful for others working on similar problem. The prior work is largely complete.

Significance:
I feel that the proposed models need to be properly evaluated before making any statements about their superiority and usefulness over previous techniques. However, assuming that they are validated correctly and shown to work well, then this work can potentially have a significant impact in NLP community where deep learning is only starting to show its usefulness.

Summary: I feel that the proposed models need to be properly evaluated before making any statements about their superiority and usefulness over previous techniques. However, assuming that they are validated correctly and shown to work well, then this work can potentially have a significant impact in NLP community where deep learning is only starting to show its usefulness.

Submitted by Assigned_Reviewer_24

This paper proposed two convolutional networks for matching text data. The main contribution can be regarded as applying the popular 2D convolutional networks to so called learning-to-match problem where each component to be matched is 1D respectively. The learning objective is based on the large margin of positive paired data to negative paired data.

The paper is discussion rich though some arguments sound not very convincing to me. However, the experimental results are very impressive. I have a few comments.

Major comments:

First, examples of LTM problems are given in the first paragraph, including machine translation. I cannot agree more that machine translation is a matching problem but why there is no such experiment? Does this mean the proposed method still cannot handle this problem? I am looking forward to seeing some results of this kind of challenging experiments.

Second, the discussion about the order-preservation property seems contradictive with the DeepMatch paper last NIPS in which the authors argued that the patch for text data cannot be same as that in image data due to the nature of text data. Can you comment on this point?

Third, in the discussion about the order-preservation property and in the discussion of the experimental results, the negative data can be constructed by randomly shuffling the words, but in all three experiments the negative data were constructed in other ways. In particular, the negative data were selected to be similar to the positive data in 5.2 but this was not mentioned in 5.3. Would the authors like to say something more on the construction of the negative data for training?

Minor comments:

As far as I know, the Siamese architecture was proposed no later than 1993. The paper was entitled ``signature verification using a "Siamese" time delay neural network'' in NIPS'93.
Summary: Strong application paper. The experimental results are impressive.

Submitted by Assigned_Reviewer_34

The paper considers text matching problems, i.e. problems involving a classification decision regarding a pair of natural language sentences (passages, texts, …).

I find the paper quite interesting (though not necessarily ground-breaking) as it provides a natural extension to some of the recent work on NN models for processing texts and images to the bi-text setting. However, I have some concerns about the evaluation (baselines / datasets).

Comments:

1) The tasks used to evaluate the model are not particularly well established, why not consider, e.g., textual entailment (RTE), QA, etc problems (not to mention MT reranking)? With these experiments, it is hard to understand how well these models actually work. Though the results seem to suggest that the proposed architecture is preferable to a few alternative NN approaches, it is not at all clear that better results cannot be obtained using (still) more conventional approaches: e.g., the models relying on inducing word / phrase alignments?

2) The way RAEs of Socher et al (NIPS 2011) are used in the submission seems quite different from how this was done in the original paper. Here, the sentences are compressed down to single vectors and an MLP is applied to predict the classification decisions. Conversely, Socher et al. (2011) used a form of pooling (over all pairs of syntactic constituents in both sentences). Their results on the MSRP (Acc 73.6/ F1 83.6) seem to be considerably better than the results reported in the submission (69.9 / 80.9). This needs to be discussed in the paper.

3) One of the fairly successful takes on using NNs in text2text applications seems much simpler (at least conceptually), as it consists of using “source” side texts when predicting target side texts with a form of NN language models (maybe relying on word alignments to define neighbourhoods as in Devlin et al. or maybe even treating the “source” side as a bag of words). It would be interesting to see how such approaches would compare with the proposed 2D compositional models.

4) A more detailed discussion of how this work is related to previous work on using convolutional models in NLP (esp. Kalchbrenner et al. 14) would be a plus. There are a few design choices here which are clearly different (e.g., the pooling function), so it would be interesting to understand the rationale (or at least to see the differences more clearly).

5) It would be nice to see a more extended discussion of what kind of compositions the model is not able to capture (e.g., this may be especially interesting in the cross lingual context or for more free order languages than English).

6) In section 4, it is mentioned that word2vec embeddings are estimated on training sets specific to each task. The paraphrase dataset seem really small (e.g., ~ 4,000 sentences in the paraphrase dataset), it does not seem realistic (?) to learn a meaningful 50-dimensional representation from these datasets. Though it may not be fair to some of the baselines, it would be interesting to see what happens if out-of-domain word representations are used (e.g., estimated for the entire wikipedia). Or, maybe, this is what has actually been done?
Summary: An interesting model, a well-written paper but not entirely convincing evaluation
Author Feedback
Author rebuttal: Thank you all for the valuable comments. About the tasks in experiment:

1. We will soon publish the dataset in Expt1 as a challenge on matching problems in NLP, in a way similar to MS sentence completion challenge.

2. The task in Expt2 is actually the key component of real-world social chatbots, e.g. that one discussed in Ref[25] or Microsoft Xiaobing (http://globalvoicesonline.org/2014/06/26/xiaobing-microsoft-chatbot-china-weibo/)

To Assigned_Reviewer_1:
1. About learning the word-embedding: We thank the reviewer for this insightful comment, and we agree that “nothing stops us from fine-tuning the embeddings”. We however beg to differ on the following 2 points:
a) This paper focuses on demonstrating the compositional property of the convolutional sentence/matching model, which is clearest when comparing to other composition models based on the same word-embeddings as building blocks. Actually it is not uncommon to evaluate “composition” model without further tuning the embeddings in supervised mode, e.g., Ref[18, 19, 21], to name a few.
b) Our other experiments (omitted for presentational simplicity) show that fine-tuning word-embedding indeed helps for most models (all except URAE+MLP and DeepMatch) to some level, while it doesn’t change the ranking of model performance. That said, we are considering including those results in the final version for a more comprehensive evaluation.

2. About Ref[9]: thanks for the suggestion. We will have more discussion on that, but Ref[9] is a classification model and cannot be directly used for matching. That said, the convolution-pooling design can be used to enrich the architecture of the two proposed architectures.

3. We didn’t have numbers of the state-of-the-arts in Table 3 solely because the table gets too long if we include all 7 results in Ref[19]. Instead, we chose to admit that ours is much lower than the best performer. We will add the numbers in the text in the final version to make it clearer.

To Assigned_Reviewer_24:
1. Our model (Architecture-II) is far more than just applying 2D CNN to matching problems. By representing the interaction between structured objects with 2D convolutions, it can model the matching patterns while maintaining the intrinsic structures of the objects at different scales. This can be readily extended to matching objects with structures of different dimension (eg. 2D in images) by using even higher-dimensional convolution. To our knowledge, this work is the first effort in that direction.

2. DeepMatch (Ref[12]) builds on bag-of-words input, which is why the “patches” (essentially sets of words) are incapable of assuming any meaningful order. Our model is fundamentally different: it feeds on natural sentences with its full sequential structure. With its order-preserving property, it models both the words and their natural order in sentences with layers of compositions.

3. About M.T reranking: We agree that it is an interesting test-bed for our model. We didn’t have experiments on that mainly because it is not easy to collect high quality data (ideally millions instances of bilingual pairs with “hard” negatives). We will be extremely interested in seeing some other group in M.T. tries this out.

4. We use randomly shuffled words as negatives merely to study the “order preserving” property of our model. In our experiments (Section 5), we use the sentences with correct orders (but “incorrect” in matching) as negatives. For task in 5.2, using just random negative is too easy, on which simple baseline achieve ~80% accuracy in 1v1 setting. We made it harder by choosing negatives similar to the positive, following a sentence completion setting in an English exam. For the task in 5.3, , random sampling already yields interesting enough negatives.

To Assigned_Reviewer_34:
On 1) : As explained in Section 5.4, our models are generic and require much more instances (>100K) than that in the current paraphrase or RTE dataset (order of 10K). This also explains why our model doesn’t achieve state-of-the-art on Expt3. About word/phrase alignment-based models: We use DeepMatch (Ref[12]) as a representative baseline in this group, since it first learns soft alignment between words and uses this as features for later matching function. Other simple models (e.g., IBM models) don't work well on Expt1 & 2 due to the great heterogeneity of the two sides and their futility to capture long-range correlation. We didn’t include them in the consideration that they are not competitive models on those tasks, but we will add them in the final version just for a more comprehensive evaluation.

On 2): The model in Socher et al (NIPS 2011) builds on the Euclidean distances between vectors for “segments” of two sentences, therefore can be applied exclusively to computing the semantic similarity between sentences (as in paraphrase identification in Expt3). It is not suitable for tasks like matching clauses (Expt1) or tweet-comment (Expt2), where we go way beyond similarity.
in Expt1&2, to make a fair evaluation of unfolding-RAE (on representing sentences as fixed-length vectors), we use a MLP to model more general matching patterns.

On 3) : We assume you are referring to the ACL14 best paper. Yes, our model is related to text2text generative models in a fundamental way. A fair empirical comparison to them would be interesting, although tricky to design (since they are not optimized for matching).

On 4) and 5): both great suggestions, we will try to add some discussion like that in the final version.

On 6): Our apology for the confusion caused by our writing. The embedding for English words (Expt1 & 3) is learnt on Wiki (~1B words), while embedding for Chinese words is learnt on Weibo data (~300M words). Both are learnt with Word2Vec in an unsupervised manner.